# An empirical method to improve rainfall estimation of dual polarization radar using ground measurements

Jungsoo Yoon<sup>1</sup>, Mi-Kyung Suk<sup>1</sup>, Kyung-Yeub Nam<sup>1</sup>, Jeong-Seok Ko<sup>1</sup>, Hae-Lim Kim<sup>1</sup>, Jong-Sook Park<sup>1</sup>

<sup>1</sup>Weather Radar Center, Korea Meteorological Adimistration

Correspondence to: Jong-Sook Park (jspark9957@gmail.com)

Abstract. This study presents an easy and convenient empirical method to optimize polarimetric variables and produce more accurate dual polarization radar rainfall estimation. Weather Radar Center (WRC) in Korea Meteorological Administration (KMA) suggested relations between polarimetric variables ( $Z - Z_{DR}$  and  $Z - K_{DP}$ ) based on a 2-D Video Distrometer (2DVD)

- measurements in 2014. Observed polarimetric variables from CAPPI (Constant Altitude Plan Position Indicator) images composed at 1.5 km of height were adjusted using the WRC's relations. Then dual polarization radar rainfalls were estimated by six different radar rainfall estimation algorithms, which are using either *Z*, *Z* and *Z*<sub>DR</sub>, or *Z*, *Z*<sub>DR</sub> and *K*<sub>DP</sub>. Accuracy of radar rainfall estimations derived by the six algorithms using the adjusted variables was assessed through comparison with raingauge observations. As a result, the accuracy of the radar rainfall estimation using adjusted polarimetric variables has
- improved from 50 % to 70 % approximately. Three high rainfall events with more than 40 mm of maximum hourly rainfall were shown the best accuracy on the rainfall estimation derived by using Z,  $Z_{DR}$  and  $K_{DP}$ . Meanwhile stratiform event was gained better radar rainfalls estimated by algorithms using Z and  $Z_{DR}$ .

# **1** Introduction

Dual polarization radars have been believed that they could provide better rainfall estimation and many researchers have 20 developed algorithms to produce better radar rainfall estimation using polarimetric variables. Seliga and Bringi (1976) derived drop size distribution from  $Z_{DR}$ , then demonstrated that rainfall can be estimated by  $Z_{DR}$ . Others proved that the rainfall estimated by using  $Z_{DR}$  was better than it was estimated based on conventional Z-R relation (Seliga et al., 1986; Aydin et al., 1987). Also, Humphrise (1974) showed that specific differential phase ( $K_{DP}$ ) was linearly related to the rain rate, and rainfall estimation algorithms using  $K_{DP}$  were suggested by other researchers (Jameson, 1985; Sachidananda and Zrnić,

1986; Chandra et al, 1990). Later Jameson (1991) suggested a rainfall estimation algorithm using both  $Z_{DR}$  and  $K_{DP}$ . Ryzhkov and Zrnić (1995a) has found that algorithms using  $Z_{DR}$  and  $K_{DP}$  were better than other algorithms. CSU (Colorado State University) algorithm and JPOLE (Joint POLarization Experiment) algorithm are synthesized methods to selectively use *Z*,  $Z_{DR}$  and  $K_{DP}$  with respect to a range of polarimetric variables or rain rate (Ryzhkov et al., 2003; Cifelli et al., 2011).

Among polarimetric variables, Z, as a backscattered variable, can be affected by attenuation or partial beam blockage. Due

to this feature, radar rainfall estimation calculated from Z is generally less than raingauge rainfall measurements (Austin, 1987; Ryzhkov and Zrnić, 1995b).  $Z_{DR}$  the ratio between horizontal reflectivity and vertical reflectivity, is related to axis ratio of hydrometeors (Zrnić and Ryzhkov, 1999; Straka et al., 2000; Frech and Steinert, 2015). When the radar antenna points vertically upward, since the shape of hydrometeors is circled toward the direction of the radar beam,  $Z_{DR}$  has to be 0

5 dB. Gorgucci et al. (1999) suggested  $Z_{DR}$  calibration from this concept.  $K_{DP}$ , defined by a range derivative of differential phase shift ( $\Phi_{DP}$ ), is propagation variable and not affected by attenuation or partial beam blockage (Ryzhkov and Zrnić, 1995a; Zrnić and Ryzhkov, 1999).  $K_{DP}$  can offer better accuracy on estimating high rain rates (Sachidananda and Zrnić, 1986; Chandarsekar et al., 1990).

It was widely accepted that three polarimetric variables, Z,  $Z_{DR}$  and  $K_{DP}$ , are related each other (Leitao and Watson, 1984; 10 Aydin et al., 1986; Ryzhkov and Zrnić, 1996; Straka et al., 2000). Straka et al. (2000) classified the hydrometeors with respect to the domains of polarimetric variables on  $Z - Z_{DR}$  space and  $Z - K_{DP}$  space. Scarchilli et al. (1996) suggested Zcalibration from self-consistency of Z,  $Z_{DR}$  and  $K_{DP}$ . Weather Radar Center (WRC) in Korea Meteorological Administration (KMA) has also suggested relations between polarimetric variables, such as  $Z - Z_{DR}$  relation and  $Z - K_{DP}$  relation, using observation measurement in order to calibrate Z and  $Z_{DR}$  of its testbed radar (WRC, 2014).

15 This study presents an easy and convenient empirical method to optimize polarimetric variables and produce more accurate dual polarization radar rainfall estimation. Above all, polarimetric variables were adjusted using the WRC's relations and assessed the accuracy of radar rainfall estimation using the adjusted variables. Then six radar rainfall estimation algorithms were applied to estimate dual polarization radar rainfalls.

This paper is composed of five sections including the introduction and conclusions. The second section is dealt with 20 applied data. The third section is explained the empirical method to improve dual polarization radar rainfall estimation. Examinations on adjusting polarimetric variables and assessing the accuracy of the radar rainfall estimation according to the adjustment are presented in the fourth section. Measurements of Yong-In Testbed (YIT) radar and rainfall records observed by AWS (Automatic Weather Station)s located within 100 km of the YIT Radar range were applied for this study.

#### 2 Data

30

25 The YIT Radar is used for this study. The YIT Radar was installed in July, 2014 as testbed radar of WRC. The YIT Radar is S-band dual polarization radar manufactured by Enterprise Electronics Corporation (EEC) and the specifications are summarized in the Table 1. WRC has been experimentally operated the YIT Radar with seasonally different scan strategy and it is still in the process of calibrating and optimizing the polarimetric variables for the YIT Radar measurements.

Primarily input data was CAPPI (Constant Altitude Plan Position Indicator) composed at 1.5 km in height from the YIT Radar. The resolution of the CAPPI is five minute interval in time and 1 km in space. Hourly accumulated rainfalls observed by 239 raingauges within 100 km of the radar range were applied to assess the accuracy of radar rainfall estimations (Fig. 1). Statistically, each raingauge covers about 131 km<sup>2</sup> and interval between two raingauges is approximately 11.4 km in distance.

Four storm events were considered in this study, as summarized in the Table 2. Event 1 was mostly stratiform

precipitation influenced by the typhoon 'Chanhom' which was mainly travelled over the West sea of South Korea and hit mainland China. It was recorded the maximum hourly gauge rainfall was just 18.0 mm during the Event 1. Other three events were frontal or convective precipitation occurred by Changma, local name of Asian summer monsoon. The maximum hourly rainfall records are 57.5, 46.0 and 77.0 mm for Event 2, 3 and 4, respectively.

# 5 **3. Empirical method**

The empirical method used in this study is purposed of improving dual polarization radar rainfall estimation through adjusting observed polarimetric variables to be on the 'Polarimetric Variables Relations' using the eleven magnitudes of adjustment based on relations of polarimetric variables (Fig. 2). In each adjustment process, radar rainfalls will be estimated by six algorithms using adjusted polarimetric variables. Then the accuracy of the each radar rainfall estimation is assessed.

10 When all the adjustment processes were completed, the polarimetric variables shown the best accuracy on the radar rainfall estimation will be taken as optimized polarimetric variables.

# 3.1 Relations between polarimetric variables derived by 2DVD

WRC built a ground observation station in Jincheon (hereafter Jincheon station). This station is aimed to verify the polarimetric variables obtained from the YIT Radar as well as radar rainfall estimation calculated using the polarimetric variables. A number of observation devices are installed in the Jincheon station; there are two weighing raingauges with resolution of 0.1 mm, six tipping bucket raingauges with three in resolution of 0.1 mm and others in resolution of 0.2 mm, two Particle Size Velocity (PARSIVEL) disdrometers and a 2-D Video Disdrometer (2DVD). Especially, 2DVD measurements are primarily used to compare with the polarimetric variables obtained from the YIT Radar and the radar rainfall estimated by using the polarimetric variables. WRC (2014) reported that the rainfall estimation calculated by the 2DVD show deviation of 5.74 % from rainfalls measured by raingauges in Jincheon station for 22 storm events occurred in 2014. In addition, two relations between polarimetric variables of the 2DVD, *Z* - *Z<sub>DR</sub>* relation (Eq. (1)) and *Z* - *K<sub>DP</sub>* relation

$$Z_{DR} = 0.153 \times Z^{0.205} \tag{1}$$

$$K_{DP} = 1.853 \times 10^{-4} \times Z^{0.781} \tag{2}$$

25 where, the units of variables are Z in  $mm^6/m^3$ ,  $Z_{DR}$  in dB and  $K_{DP}$  in  $^{\circ}/km$ .

The two relations are depicted as black solid line in the Fig. 3 and entitled as 'Polarimetric Variables Relations'. These are reasonable as they were shown in the range of the rain domains suggested by other researches (Vivekanandan et al., 1999; Straka et al., 2000). Thus these relations are taken as referential relations in this study.

#### 3.2 Adjustment of polarimetric variables using the relations

30 The bivariate distributions of  $Z - Z_{DR}$  and  $Z - K_{DP}$  observed by the YIT Radar are determined as plotted hatched area on the

10

Fig. 2(a). The modes of the observed bivariate distributions will be adjusted to be in the dashed polygon on the 'Polarimetric Variables Relations' (Fig. 2(a)). It is, however, uncertain where the adjusted modes would be on the line of the relations along the adjustment processes. In case of Z without bias, the modes will be adjusted with vertical shift along the Y-axis. In other words,  $Z_{DR}$  and  $K_{DP}$  will be either increased or decreased without adjustment of Z until the observation modes will be set on the 'Polarimetric Variables Relations'.

5

In other case where Z has bias, the bias in Z can be varied due to environmental interferences, such as temperature or humidity, which will have impact on radar performances and measurements. Therefore it has to be considered the degree of adjustment. Eleven levels of adjustment magnitude are set in the range of 0 to 10. For each magnitude level, Z was increased from 0 dBZ to 10 dBZ with interval of 1 dBZ. Level '0' means no bias in Z. Then the  $Z_{DR}$  and  $K_{DP}$  were increased or decreased in order to the mode of the observed bivariate distributions to be on the 'Polarimetric Variables Relations'.

Polarimetric variables were adjusted using the eleven magnitudes of adjustment for the relations of  $Z - Z_{DR}$  and  $Z - K_{DP}$ for the four events. Table 3 is summarized the adjusted results of Event 4 to show how the magnitude of adjustment are set for the event, as an example. The modes of observed  $Z - Z_{DR}$  and  $Z - K_{DP}$  distribution are 25.750 - 0.350 and 44.750 - 1.050, respectively. With regarding to the Magnitude 5, the polarimetric variables Z,  $Z_{DR}$  and  $K_{DP}$  have to be increased 5 dBZ, 0.303

dB and 0.373 °/km to the observed Z,  $Z_{DR}$  and  $K_{DP}$ , respectively in order to adjust modes to be on the relations. 15

### 3.3 Radar rainfall estimation and assessment of accuracy

In order to validate the empirical method based on the WRC's relations, six radar rainfall estimation algorithms were hired for this study, as summarized in the Table 4.

The algorithm,  $R_1$ , is considered as most popularly used method by hydrologist and KMA in Korea (Yoo et al., 2016). It is a conventional estimation algorithm (Marshall and Palmer, 1948). As this algorithm was derived from the cases of stratiform 20 precipitation, it has been shown underestimation for high rainfalls (Battan, 1973; Ryzhkov and Zrnic, 1996).

Three algorithms calculate radar rainfall estimation using two polarimetric variables, Z and  $Z_{DR}$ . R2 was introduced by Bringi and Chandraseker (2001), R3 was derived by Brandes et al. (2003), and R4 was suggested by WRC (2014) using 2DVD measurements for 22 summer storm events.

- 25 The other two algorithms are using three polarimetric variables, Z,  $Z_{DR}$  and  $K_{DP}$ . The algorithm R5 was suggested by Colorado State University and named CSU-ICE algorithm (Cifelli et al, 2011). R5 is varied with four different algorithms, R(Z),  $R(Z, Z_{DR})$ ,  $R(K_{DP}, Z_{DR})$  and  $R(K_{DP})$  at the given ranges of the three polarimetric variables. R6 is named as JPOLE algorithm, which uses three equations to estimate the radar rainfall depending on the rainfall intensity as presented on the Table 4 (Ryzhkov et al., 2003).
- The accuracy of the radar rainfall estimation derived by the adjusted polarimetric variables at each process was assessed 30 using the Eq. (3). As values are getting closer to '100 %', the radar rainfall estimation is getting better. As 1-NE (Normalized Error) quantifies the absolute error, maximum 1-NE is minimized errors for both bias and random error.

$$1 - \mathrm{NE} = \left(1 - \frac{\sum |R_i - G_i|}{\sum G_i}\right) \times 100(\%) \tag{3}$$

where,  $R_i$  is the radar rainfall (mm) for the *i*-th data pair,  $G_i$  is the gauge rainfall (mm) for the *i*-th data pair.

The accuracy of the radar rainfall estimation, gained from the adjusted polarimetric variables, was calculated in the comparison with rainfall records of the raingauges. As precipitation echoes are moving and wind impacts on the direction of falling raindrops towards ground, it is not easy to have exactly matched value for each grid (1 km × 1 km) with rainfall record of the raingauge. With this reason, rainfall estimation values of a centroid grid and eight surrounded grids were compared to the rainfall record of the raingauge (Fig. 4). Among the nine values, the best matched value was taken accounted into the calculation of the rainfall estimation accuracy.

# 4 Results

- The accuracy of the radar rainfall estimations derived by six algorithms (Table 4) was gained by Eq. (3) as illustrated in Fig. 5. The accuracy of the rainfall estimation using observed polarimetric variables without adjustment is plotted along the Y-axis, named as 'No Adj.'. Generally plotting diagrams show increasing accuracies as level of adjustment is getting higher from M0 to the magnitude of adjustment, which gives the best accuracy. Once each event reaches the best accuracy with certain magnitude of adjustment on the polarimetric variables, the accuracy is getting less accurate with increasing level of adjustment.
- adjustment.

Event 1 is almost symmetrically changed before and after showing the best accuracy with M5. Two convective events (Event 2 and 4) form a sort of plateau with gradually increased longer upside hill as shown diagrams Fig. 5(b) and (d), as there is no big changes after M7 which shows best accuracy. Event 3 is quite similar like Event 1, although wider range of decreasing rate on accuracies for the area of M6 – 10. Nevertheless, six algorithms in all events commonly show a certain magnitude of adjustment produced the best accuracy of the rader minfolls estimation.

magnitude of adjustment produced the best accuracy of the radar rainfalls estimation.

Table 5 is summarized the accuracy gained by six algorithms using observed ('Before') and adjusted ('After') polarimetric variables. The accuracy of the radar rainfall estimation using adjusted polarimetric variables shows more than 70 % of accuracy for most cases. Event 3 is even reached more than 80 % of accuracy by all algorithms using adjusted polarimetric variables.

Stratiform precipitation (Event 1) shows accuracies from 66.9 % to 71.9 % for the rainfalls estimated by all algorithms. The *R*5 and *R*6 are, actually, hardly treated  $K_{DP}$  for Event 1, therefore the performances are similar to the accuracy shown by other algorithms using *Z* and *Z*<sub>DR</sub> (*R*2, *R*3 and *R*4). Convective precipitations (Event 2, 3 and 4) gain better accuracies on rainfall estimation derived by either *R*5 or *R*6.

The algorithm, *R*4, suggested by WRC fairly performed for the four events. Best accuracy for each event was gained using 30 polarimetric variables adjusted with magnitude of 5 (Event 1 and 3) and 8 (Event 2 and 4). In the category of algorithms using *Z* and *Z*<sub>DR</sub>, *R*4 shows very similar performances with *R*2 and *R*3, although each algorithm gets the best accuracy using

polarimetric variables adjusted by different magnitudes. Thus the relations of polarimetric variables applied for the *R*4 can be regarded as suitable to use for estimating radar rainfalls of the YIT Radar.

Figure 6 is comparatively illustrated hourly rainfalls observed by raingauge and estimated by six algorithms using the polarimetric variables adjusted with best performed magnitudes, as given in Table 4, for the Event 4. Fig. 6(a) is scatter

- 5 diagram of hourly gauge rainfalls and radar rainfall estimation derived from polarimetric variables without adjustment. All algorithms produced underestimated radar rainfall estimations, particularly for rainfall records greater than 20mm/hr. The maximum raingauge record 77.0 mm for the event was estimated as just 19.7 mm by *R*1 which is quantitatively 25.6 % of the raingauge records. Meanwhile, the radar rainfall estimations gained by the algorithms (*R*2, *R*3 and *R*4) of using *Z* and  $Z_{DR}$  are shown slightly better relationship with observed data than by *R*1, although they are still underestimated. Before adjusting
- 10 polarimetric variables the algorithms *R*5 and *R*6 have far better relationship as appeared in the Fig. 6(a). The maximum radar rainfalls estimated by *R*5 is 51.4 mm and is quantitatively about 66.8 % of the maximum raingauge record. Generally all six algorithms have much improved relationship between gauged and estimated rainfalls using polarimetric variables adjusted by the best suited magnitudes as plotted in Fig. 6(b). The radar rainfall estimations derived by R5 shows the most well matched relationship with observed data.
- This can be reassured with Fig. 7 and 8. As the algorithm *R*5 shows overall better rainfall estimations, the observed bivariate distributions of  $Z Z_{DR}$  and  $Z K_{DP}$  of the four events are plotted in Fig. 7(a) and (c), respectively. The black solid lines in the Fig. 7 are presented the 'Polarimetric Variables Relations' between *Z* and  $Z_{DR}$  of Eq. (1) and  $Z K_{DP}$  of Eq. (2). The adjustment results of *R*5 with given magnitudes are presented in Fig. 7(b).

In the Fig 7(a), all events have areas where the observed bivariate distributions between Z and  $Z_{DR}$  are not correspondent with the referential relations (solid line in each figure). These areas were move to be on the referential relations line after

- with the referential relations (solid line in each figure). These areas were move to be on the referential relations line after adjusted with best performed magnitudes for observed polarimetric variables. For the *R*5, the best suited adjustment are gained with magnitude of M5 for Event 1, M7 for Event 2, M3 for Event 3 and M7 for Event 4 as shown in Fig. 7(b). These levels of magnitudes mean adding 5, 7, 3 and 7 dBZ to the observed *Z* of Event 1, 2, 3 and 4, respectively, at the same time  $Z_{DR}$  was increased in the relation of *Z*.
- With regarding to the bivariate distribution of  $Z K_{DP}$  as depicted in Fig. 7(c) and (d), Event 1 is not analyzable of  $K_{DP}$ , as the maximum hourly rainfall is 18 mm and Z is less than 38 dBZ which are least values to give meaningful  $K_{DP}$  (Table 4). As presented in Fig. 7(d), the adjustment results with M7 for Event 2 and 4 and M3 for Event 3 lead the high probability (40 

5

30

algorithms to estimate radar rainfalls. The magnitude of 5 is actually high, as it means the observed reflectivity is lower than 5dBZ in order to produce quite closed amount of rainfalls to the measurements by raingauges. It is guessed that, as mentioned earlier, the YIT Radar was installed in July, 2014 and is still required to calibrate its polarimetric variables. Even if the YIT Radar is well calibrated, it is expected that the accuracy of the radar measurement have to be improved with suspicion of systematic bias caused by  $Z - Z_{DR}$  relation and  $Z - K_{DP}$  relation. To overcome the bias, it can be suggested that the relation has to be derived using longer term measurements for more various precipitation events. Finally, the attenuation of Z due to the partial beam blockage can cause the error. Since Z is very low in the region of the partial beam blockage, it can make the magnitude high.

#### **5** Conclusions

- 10 This study has reviewed an easy and convenient empirical method to optimize polarimetric variables and produce more accurate dual polarization radar rainfall estimation. Above all, observed polarimetric variables were adjusted using the WRC's relations of polarimetric variables ( $Z - Z_{DR}$  and  $Z - K_{DP}$ ). Radar rainfall was estimated by six algorithms, and its accuracy was assessed for the four summer rainfall events occurred in 2014. The results are as follow.
- First, accuracies increased as level of adjustment was getting higher from the M0 to the magnitude of adjustment, which gave the best accuracy. Once each event reached the best accuracy with certain magnitude of adjustment on the polarimetric variables, the accuracy was getting less accurate with increasing level of adjustment. Six algorithms in all events commonly show a certain magnitude of adjustment produced the best accuracy of the radar rainfalls estimation.

Second, the observed bivariate distributions between polarimetric variables were not correspondent with the referential relations. These distributions were move to be on the referential relations line after adjusted with best performed magnitudes

20 of adjustment for observed polarimetric variables. For example, for the *R*5, the best performed magnitudes were M5 for Event 1, M7 for Event 2, M3 for Event 3 and M7 for Event 4.

Third, the accuracy of the radar rainfall estimation using adjusted polarimetric variables showed almost more than 70 % of accuracy for most cases. Event 3 was even reached more than 80 % of accuracy. Three high rainfall events with more than 40 mm of maximum hourly rainfall were shown the best accuracy on the rainfall estimation using Z,  $Z_{DR}$  and  $K_{DP}$  (*R*5 and

25 *R*6). Meanwhile, the accuracy of the radar rainfall estimated by *R*4 suggested by WRC was almost similar to the accuracy of other algorithms using *Z* and  $Z_{DR}$  (*R*2 and *R*3). Therefore the relations of polarimetric variables applied for the *R*4 can be regarded as suitable to use for estimating radar rainfalls of the YIT Radar.

Overall, unmatched radar rainfall estimation compared to observed rainfalls has been much improved by six rainfall estimation algorithms combined with the empirical method for the four cases. Although, there are still underestimated radar rainfalls which were caused by the attenuation of Z due to the partial beam blockage still existed.

Through this study, the empirical method to adjust polarimetric variables using the referential relations suggested by WRC proved that it is a reliable method to overcome biases on measurements by dual polarization radars in order to estimate radar rainfalls. It will be useful to quantitatively improve the radar rainfall estimation of newly install radars, as to establish

optimal or reliable quality control algorithms on new radars is taken long time, such as YIT Radar. In addition, the empirical method can be useful to improve the accuracy of radar rainfall estimations derived by different algorithms through adjusting polarimetric variables. Nevertheless there are still long way to improve the method when it applies for, particularly radar measurements with partial beam blockage and severe systematic biases. Thus this method will be continued to develop through applications with more various precipitation types and try to adjust the polarimetric variables in real time in near

# Acknowledgement

This research was supported by the "Development and application of cross governmental dual-pol radar harmonization (WRC-2013-A-1)" project of the Weather Radar Center, Korea Meteorological Administration.

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

# 15 Meteor., 25, 1475-1484, 1986.

Austin, P. M.: Relation between measured radar reflectivity and surface rainfall, Mon. Weather.Rev., 115, 1053-1070, 1987. Battan, L. J.: Radar observation of the atmosphere, University of Chicago Press, 324pp., 1973.

- Brandes, E. A., Zhang, G., and Vivekanandan, J.: An evaluation of a drop distribution-based polarimetric radar rainfall estimator, J. Appl. Meteorol., 42, 652-660, 2003.
- 20 Bringi, V. N. and Chandarsekar, V.: Polarimetric Doppler weather radar: Principles and applications, Cambridge University Press, 636 pp., 2001.
  - Chandrasekar, V., Bringi, V. N., Balakrishnan. V. N., and Zrnić, D. S.: Error structure of multiparameter radar and surface measurements of rainfall. Part III: Specific differential phase, J. Amos. Oceanic. Technol., 7, 621-629, 1990.

Cifelli, R., Chandrasekar, V., Lim, S., Kennedy, P. C., Wang, Y., and Rutledge, S. A.: A New Dual-Polarization Radar Rainfall Algorithm: Application in Colorado Precipitation Events, J. Atmos. Oceanic Technol., 28, 352-364, 2011.

Frech, M. and Steinert, J.: Polarimetric radar observations during an orographic rain rate, Hydrol. Earth. Syst. Sci., 19, 1141-1152, 2015.

Gorgucci, E., Scarchilli, G., and Chandrasekar, V.: A procedure to calibrate multiparameter weather radar using properties of the rain medium, IEEE. T. Geosci. Remote., 37, 269-276, 1999.

30 Humphries, R. G.: Depolarization effects at 3 GHz due to precipitation. Storm Weather Group Scientific Report MW-82, McGill University, Montreal, Quebec, 84pp., 1974.

10

Jameson, A. R.: Microphysical interpretation of multi-parameter radar measurements in rain. Part III: Interpretation and measurement of propagation differential phase shift between orthogonal linear polarizations, J. Atmos. Sci., 42, 607-614, 1985.

Jameson, A. R.: A comparison of microwave techniques for measuring rainfall, J. Appl. Meteor., 30, 32-54, 1991.

5 Leitao, M. J. and Watson, P. A.: Application of dual linearly polarized radar data to prediction of microwave path attenuation at 10–30 GHz, Radio. Sci., 19, 209–221, 1984.

Marshall, J. S. and Palmer, W. M.: The distribution of raindrops with size, J. Meteorol., 5, 165-166, 1948.

Mason, B. J.: The Physics of Clouds, 2d ed. Oxford University Press, 671 pp., 1971.

Ryzhkov, A. V., Giangrande, S. E., and Schuur, T. J.: Rainfall measurements with the polarimetric WSR-88D Radar, NOAA/NSSL Report, 98 pp., 2003.

Ryzhkov, A. V. and Zrnić, D. S.: Comparison of dual polarization radar estimators of rain, J. Amos. Oceanic. Technol., 12, 249-256, 1995a.

Ryzhkov, A. V. and Zrnić, D. S.: Rain in shallow and deep convection measured with a polarimetric radar, J. Atmos. Sci., 53, 2989-2995, 1996.

Scarchilli, G., Gorgucci, E., Chandrasekar, V., and Dobaie, A.: Self-consistency of polarization diversity measurement of

- rainfall, IEEE. T. Geosci. Remote., 34, 22-26, 1996.
  - Seliga, T. A. and Bringi, V.N.: Potential use of radar differential reflectivity measurements at orthogonal polarization for measuring precipitation, J. Appl. Meteorol., 15, 69-76, 1976.
  - Seliga, T.A., Aydin, K., and Direskeneli, H.: Disdrometer measurements during an intense rainfall event in Central Illinois Implications for differential reflectivity radar observations, J. Climate. Appl. Meteor., 25, 835-846, 1986.
- Straka, J. M., Zrnić, D. S., and Ryzhkov, A. V.: Bulk hydrometeor classification and quantification using polarimetric radar data: synthesis of relations, J. Appl. Meteorol., 39, 1341-1372, 2000.
  - Vivekanandan, J., Zrnić, D. S., Ellis, S. M., Oye, R., Ryzhkov, A., and Straka, J.: Cloud microphysics retrieval using S-band dual-polarization radar measurements, Bull. Amer. Meteor. Soc., 80, 381-388, 1999.

WRC (Weather Radar Center): Development and application of cross governmental dual-pol. radar harmonization., Seoul,

- Yoo, C., Yoon, J., Kim, J., and Ro, Y.: Evaluation of the Gap-Filler with respect to the quality of radar data in Korea, Meteorol. Appl., DOI: 10.1002/met.1531, 2016.
- Zrnić, D. S. and Ryzhkov, A. V.: Polarimetry for weather surveillance radars, Bull. Amer. Meteor. Soc., 80, 389–406, 1999.

Ryzhkov, A. V. and Zrnić, D. S.: Precipitation and attenuation measurements at a 10-cm wavelength, J. Appl. Meteor., 34, 2121-2134, 1995b.

Sachidananda, M. and Zrnić, D. S.: Differential propagation phase shift and rainfall rate estimation, Radio. Sci., 21, 235-247, 1986.

<sup>30</sup> Korea, 268pp., 2014.

Table 1. Specifications of the YIT Radar

| Model                                           | DWSR-8501 S/K-SDP |  |  |  |  |
|-------------------------------------------------|-------------------|--|--|--|--|
| Manufacturer                                    | EEC (U.S.)        |  |  |  |  |
| Antenna diameter (m)                            | 8.5               |  |  |  |  |
| Beam width (°)                                  | 1                 |  |  |  |  |
| Transmitting tube                               | Klystron          |  |  |  |  |
| Wavelength (cm)                                 | 10.41             |  |  |  |  |
| Transmitting frequency (MHz)                    | 2,879             |  |  |  |  |
| Peak power (kW)                                 | 850               |  |  |  |  |
| Maximum / effective<br>observational range (km) | 500 / 240         |  |  |  |  |