# Peer review of "An empirical method to improve rainfall estimation of dual polarization radar using ground measurements"

_Hydrology and Earth System Sciences, 2016_

## Referee Comment (RC1) · Anonymous Referee #1 · 31 Mar 2016

**An empirical method to improve rainfall estimation of dual polarization radar using ground measurements**
**$H$ESS-2016-27**

Revision

March 31, 2016

**Manuscript revision**

The manuscript *H*ESS-2016-27 deals with the estimation of rainfall by means of polarimetric weather radar data, in particular with the optimization of polarimetric measurements from the point of view of an operational weather service.

Many weather services own polarimetric radars, but polarimetric rainfall estimates are still not currently applied to obtain operational products. The topic is therefore of interest both for the hydrological and for the remote-sensing scientific and operational communities.

I see some potential and interesting novelties in the manuscript. However, several major issues in the current version do not allow one to conduct a proper review. My recommendation would be to re-submit the manuscript after a major revision of contents and form, for a new complete revision.

The following sections list the general and specific issues of the paper, and provide some suggestions for its reorganization.

**General issues**

1. English Language: i am not a native English speaker, and i sympathise with the authors for the difficulties that all the non-native speakers encounter during the revision of a manuscript. However, i felt in this case that the quality of the language is preventing me from providing a good and helpful review.

2. Manuscript presentation: the manuscript is short. This is not necessarily a bad aspect, but the sections deserve much more details than what actually provided. I believe that this is true for all the section, but to provide some examples:

    - Sec. 2. No information is given (in the text) about the technical specification of the radar or about the (detailed) characteristics of the 4 storms. If only 4 storms have to be used, they should be described with a high level of detail in order to properly comment on the generality and robustness of the results.

    - Sec. 3.2. This is the core of the manuscript, and half a page is not enough for the reader to understand how the method works. By looking at the diagram of Fig. 2 and by reading the text, i feel that there is a good idea but i do not have enough information to fully grasp it and see its merits (or faults).

    - Sec 3.3. This section should also explain how the same set of rain-gauges is used both in the optimization and in the validation of the novel method. If no additional information is given it is not surprising that, after optimization on raingauges, the accuracy (with respect to the raingauges themselves) increases.

3. Dataset. The manuscript is based on a dataset of 4 events that show two different behaviours in the optimization (as seen in Fig. 5). This dataset, unless additional explanations are provided, is very limited: it becomes hard to generalize the results and it is difficult to explain the reason of the different behaviours of Fig. 5.

4. Scope. The manuscript is relatively technical and in my opinion it should have been submitted to a different Copernicus journal: Atmospheric Measurement Techniques (AMT).

**Major merit**

The main merit of the paper (to be clarified and better explained) is the idea to adapt the bivariate distributions of polarimetric variables by moving their centres of mass towards an expected mutual behaviour, as illustrated in Fig. 2.

**Specific issues**

1. Introduction. Some relevant literature may be helpful here, to complete the overview. I suggest Matrosov et al. (1999); Illingworth (2004); Matrosov (2010); Wang and Chandrasekar (2010).

2. Page 1, Line 25: "Chandra" should be "Chandrasekar".

3. Page 2, Line 1: $Z_{DR}$ is a ratio if $Z_H$ and $Z_V$ are expressed in linear units.

4. Page 2, Line 29: you should definitely comment on the fact that you compare ground measurements with measurements collected at much higher altitudes (1.5 km) and on the possible sources of error that comes from the microphysical processes occurring below 1.5 km.

5. Page 3, Line 8: here the term "eleven magnitudes" appears, but it has not been defined. The reader may be lost.

6. Page 3, Line 17: is the PARSIVEL used in this study? If not, he can also not be mentioned.

7. Page 3, Line 27: could you show on Fig. 3 also these relations?

8. Page 4, Line 8: why only positive magnitudes are considered?

9. Page 4, Lines 13-15: this sentence needs some visual support (a figure), to guide the reader to understand the algorithm.

10. Page 4, Line 18 (Table 4): add references for all the relations in the table.

11. Page 5, Line 18: You should comment about those 2 behaviours (maximum around 5 dB of events 1 and 3 vs asymptotic behaviour of events 2 and 4), and here it would be helpful to understand if the type of rainfall was very different in those cases.

12. Page 6, Line 25 (Fig 7): could you specify which $K_{dp}$ estimation method you employ? Sometimes the estimates of $K_{dp}$ seem poor (as in Event 1)

**Tables and Figures**

1. Provide more complete information in the caption of Figures and Tables. Captions are often too short and not complete.

2. Table 2: it is a good starting point, but the description of the events should be more detailed and supported by actual radar images (PPI or CAPPI) for each event.

3. Table 4: add a reference for the algorithms, in the same table.

4. Figure 2: i like this figure, but it needs to be explained step by step with additional details in the text.

5. Figure 4: add an indication of vertical distance between the radar measurement and the gauge.

**Bibliography**

Illingworth, A. J., 2004: Improved precipitation rates and data quality by using polarimetric measurements. *Weather radar: principles and advanced applications*, P. Meischner, Ed., Springer, 130–166.

Matrosov, S. Y., 2010: Evaluating polarimetric X-band radar rainfall estimators during HMT. *J. Atmos. Oceanic Technol.*, **27 (1)**, 122–134, doi:10.1175/2009JTECHA1318.1.

Matrosov, S. Y., R. A. Kropfli, R. F. Reinking, and B. E. Martners, 1999: Prospects for measuring rainfall using propagation differential phase in X- and $K_a$-radar bands. *J. Appl. Meteor.*, **38 (6)**, 766–776.

Wang, Y. T. and V. Chandrasekar, 2010: Quantitative precipitation estimation in the CASA X-band dual-polarization radar network. *J. Atmos. Oceanic Technol.*, **27 (10)**, 1665–1676, doi:10.1175/2010JTECHA1419.1.

---

## Referee Comment (RC2) · Anonymous Referee #2 · 1 Apr 2016

The authors suggest tweaking the measured values of Z, Zdr, and Kdp to match the average expected dependencies of Zdr and Kdp on Z or the bivariate distributions obtained from the disdrometer-based simulations. The "reference" dependencies are specified in Eqs 1 and 2. The major problem with such approach is that there are no universal reference dependencies valid for all rain types. For example, the Z – Zdr average dependency for tropical rain generated by a warm rain process is quite different from the one for continental rain which mostly originates from the ice aloft. For a given Z, Zdr in tropical rain is significantly lower than in continental rain, particularly at higher rain rates. A similar rule holds for the Z – Kdp dependency. In fact, using the suggested methodology, the authors deny the impact of the DSD variability on

the performance of radar rainfall algorithms. I guess that the improvement in the QPE performance caused by the recommended adjustment is mainly due to mitigation of the measurement biases in Z and Zdr. The description of the adjustment routine in section 3.2 is very brief and insufficient for understanding or reproducing the methodology. The adjustment of Zdr or Kdp for a given Z looks straightforward but the procedure for Z adjustment is totally unclear. Obvious underestimation of rainfall, say, by using the R(Z) relation illustrated in Fig. 6a could be caused by either negative bias in the Z measurements or by the very nature of the observed rain (e.g., tropical) for which a power-law R(Z) relation with higher intercept is required. How to distinguish between these two sources of error? A range of needed adjustment (likely attributed to negative Z bias) between 3 and 10 dB shown in Table 5 is quite disturbing because it may point to a serious problem with radar calibration. The magnitude of such bias is too high for operational weather radars. Moreover, the magnitude of the Z adjustment for a single rain event can vary by as much as 3 dB depending on the algorithm choice. To me this is an indication that both Z bias and the DSD variability (which differently affects the performance of various rainfall relations) may come into play. The English usage has to be improved dramatically since even the meaning of several sentences is "lost in translation". There is inaccurate statement regarding the methodology of Seliga and Bringi for DSD retrieval and rainfall estimation (first paragraph in Introduction). It is not a single Zdr but the combination of Z and Zdr which was proposed to address these problems. The concept of the suggested methodology is flawed and its description is insufficient and hard to understand. Therefore I recommend rejection.

---

## Author Comment (AC1) · 23 Apr 2016

**Responses to the Reviewer's Comments**

**Reviewer #1:**

**Overall Response:** The reviewer commented 4 general issues, 17 specific issues and 5 comments for tables and figures. We sincerely thank you for the comments that help to improve our paper. We will improve our manuscript as the reviewer's commented and the detail plans are as followed.

**General issues:**

**Q1:** *English Language: i am not a native English speaker, and i sympathise with the authors for the difficulties that all the non-native speakers encounter during the revision of a manuscript. However, i felt in this case that the quality of the language is preventing me from providing a good and helpful review.*

**Response:** All the authors are well aware of that our manuscript is required professional English review, and our revised manuscript will be getting a professional English correction before it will be submitted. Hope this process will improve our manuscript to reach high standard of English requirement for the journal.

**Q2.1:** *Manuscript presentation: the manuscript is short. This is not necessarily a bad aspect, but the sections deserve much more details than what actually provided. I believe that this is true for all the section, but to provide some examples:*

*Sec. 2. No information is given (in the text) about the technical specification of the radar or about the (detailed) characteristics of the 4 storms. If only 4 storms have to be used, they should be described with a high level of detail in order to properly comment on the generality and robustness of the results.*

**Response:** As recommended by the reviewer, we will give a full description about the technical specification of the radar and the characteristics of the 4 storms. For example, we gave an explanation of event 1 in the manuscript, which was mostly stratiform precipitation (below figure). We will describe each storm with additional information. Each event will be described using synoptic weather condition; distribution of rainfall on the ground based on raingauge measurement and used radar images.

**Event 1:** Event 1 was related to the Typhoon Chan-Hom, which was developed near the Equator, traveled through West Sea of Korea and finally hit mainland China (Fig. A).

Korea had light or moderate rain over the most part of the country and rain was lasted over 24 hours since late 11th July 2015(Fig. B). Observed hourly maximum rainfall was 18mm at 201507120900KST during the event. Fig. C is shown CAPPI reflectivity image composed at 1.5 km in height using data observed by the YIT radar at 9am on 12th July 2015. Black circle is represented 100km in horizontal distance from the YIT radar and only inner circled areas are used for this study. The precipitation type was mainly stratiform rain with very clearly observed bright band as supportive evidences of Fig. D(Z), E(. $\rho_{hv}$) and F(ZDR) observed by the YIT radar at $9.12^{o}$ . Bright band was developed about 4.5km in height, therefore the used CAPPI image composed at 1.5km was not influenced by bright band. The YIT radar has been purposely set a beam blockage area around 0~10degree to prevent intervenes by neighboring telecommunication radar.

[Figure]

| A. Weather surface chart @201507120900KST | B. AWS- 60min accumulated rainfall @ 201507120900KST | C. YIT CAPPI (1.5km) CZ@ 201507120900 KST |
| --- | --- | --- |
| D. CZ@ 201507120900 KST | E. $\rho_{hv}$ at 201507121000KST | E. ZDR@ 201507120900 KST |

**Event 2:** During this Event 2(23~26 July 2016), southern cold front was faced with warm front from the North in middle of Korea Peninsula and stayed over 72 hours as presented in Figure A, B and C. As strong frontal precipitation developed, the maximum hourly rainfall was recorded with 57.5mm at 2am on 25[th] July 2015 as shown Fig.D. Leading convective cells were lined from the South-West to the North-East with

stratiform rain developed surrounded area.

[Figure]

| A. Weather surface chart @201507232100KST | B. Weather surface chart @201507252500KST | C. Weather surface chart @201507262100KST |
| --- | --- | --- |
| D. Max hourly rainfall 57.5mm @ 201507250200 KST | E. CAPPI CZ at 201507250200 KST | E. Vertical profile of CZ at 201507250200 KST |

**Event 3:** This event (201507290000~2300KST) was occurred during the Asian summer monsoon.  Frontal precipitation band was developed midland in South Korea. Multi super cell storms were traveled from the West Sea and passed mainland, and light or moderate stratiform echoes were largely developed (Fig. A). Naturally heavy rain was led by storms; hourly maximum rainfall was recorded 46mm at 10am on 25th July 2015

(Fig.B).

The strong cells were activated from 8am to 11 am on the day and it was light rain rest of the period at the event. With the figure below, the processes of strong, invigorate, deep and rotating updraft are well presented on a scale of (1 ~ 20km) with trailing large stratiform at 8am on 29th July 2015. Thus this event can be categorized as a mixture of super cell storms and very light stratiform rain.

[Figure]

**Event 4:** The Event 4 is typical mid-latitude squall line system accompanied by strong lightning as presented in Fig. A and B. The linear convective cells were developed particularly from non to 19pm on 8th August 2015. During this time, persistent thunderstorms and contiguous precipitation areas were produced. The strongest reflectivity was appeared around 55dBZ, which is surrounded by 45~50dBZ. Maximum hourly rainfall was 77mm at 201508081500KST.

[Figure]

| A. 60min accumulated rainfall in mm at 201508081500KST | B. Lightning hit number for 10min at 201508081500KST |

[Figure]

| C. Used CAPPI reflectivity at 201508081500KST | D. Observed PPI reflectivity at 201508081500KST |
|---|---|

**Q2.2:** *Sec. 3.2. This is the core of the manuscript, and half a page is not enough for the reader to understand how the method works. By looking at the diagram of Fig. 2 and by reading the text, i feel that there is a good idea but i do not have enough information to fully grasp it and see its merits (or faults).*

**Response:** The entire section 3 is about the methodology and has supplementary three sub-sections, although the overall structure of the section is not well described at the beginning. We have been working on enhance the section 3 with further detail explanation and supplementary information.

**Q2.3:** *Sec 3.3. This section should also explain how the same set of rain-gauges is used both in the optimization and in the validation of the novel method. If no additional information is given it is not surprising that, after optimization on raingauges, the accuracy (with respect to the raingauges themselves) increases.*

**Response:** We hope this question will be answered with our revised manuscript, which will be amended as described in the previous suggestion relevant to the Section 3.

**Q3:** *Dataset.* *The manuscript is based on a dataset of 4 events that show two different behaviours in the optimization (as seen in Fig. 5). This dataset, unless additional expla-nations are provided, is very limited: it becomes hard to generalize the results and it is difficult to explain the reason of the different behaviours of Fig. 5.*

**Response:** As we have already worked for providing further details of the four events, this concern will be sort out. We, however, will enhance interpretation of the results in the relation of characteristics of each event. This will make this section becomes strong.

**Q4:** _Scope._ *The manuscript is relatively technical and in my opinion it should have been submitted to a different Copernicus journal: Atmospheric Measurement Techniques (AMT).*

**Response:** We had thought about this a lot, too as the reviewer pointed out. Also we agree your comment that our current version of manuscript is relatively technical. It is much complicated to estimate rainfall of using dual-pol. radar, because multivariate analysis (Z, ZDR and KDP) have to be involved in the processes. Still many other researchers or radar operators are seeking for the optimal method to improve QPE. Radar data QPE has been of more interested to hydrologist rather than any other users of weather radar product. First and correspondence authors are hydrologist, too. Since we concluded the empirical method in this study can contribute for those who have been suffered or concerned QPE with dual-pol radars. HESS is one of the most popular journals for hydrologist, and we decided to submit our paper to HESS.

**Specific issues:**

**Q1:** *Introduction. Some relevant literature may be helpful here, to complete the overview. I suggest Matrosov et al. (1999); Illingworth (2004); Matrosov (2010); Wang and Chandrasekar (2010).*

**Response:** As recommended by the reviewer, we will refer the literature as suggested in introduction.

**Q2:** *Page 1, Line 25: "Chandra" should be "Chandrasekar".*

**Response:** As the name was miswritten, we will amend the name.

**Q3:** *Page 2, Line 1: $Z_{DR}$ is a ratio if $Z_H$ and $Z_V$ are expressed in linear units.*

**Response:** As recommended by the reviewer, ZDR defines as a ratio when ZH and ZV are expressed in linear units. Also ZDR can define as differences when ZH and ZV are expressed in dB units. Since this point can confuse readers, we will correct the sentence.

**Q4:** *Page 2, Line 29: you should definitely comment on the fact that you compare ground*

*measurements with measurements collected at much higher altitudes (1.5 km) and on the possible sources of error that comes from the microphysical processes occurring below 1.5 km.*

**Response:** In fact the gap between measurement on the ground and measurement aloft also can have great effect on the error of the radar measurement. As there are many mountains, covering about 70% of the entire Korean Peninsula, the radars installed in Korea are, also, largely affected by the geomorphological feature. Nevertheless, we used 1.5 km CAPPI because the altitude (1.5 km) was the minimum height determined to secure rather homogeneous data in altitude without blocking a significant portion of terrain. We will enhance the explanation about the error due to the gap.

**Q5:** *Page 3, Line 8: here the term \eleven magnitudes" appears, but it has not been defined. The reader may be lost.*

**Response:** As recommended by the reviewer, we will define the term as suggested.

**Q6:** *Page 3, Line 17: is the PARSIVEL used in this study? If not, he can also not be mentioned.*

**Response:** We did not use PARSIVEL. We mentioned it in manuscript to explain Jincheon ground station in where various meteorological instruments including PARSIVEL are installed. Jincheon station has an important role, because the station is aimed to verify the polarimetric variables obtained from the YIT Radar as well as radar rainfall estimation calculated using the polarimetric variables. Also, the polarimetric variables can be retrieved by PASIVEL, however, we have more confidence in 2DVD when verifying the polarimetric variables. Since this point can confuse readers, we will improve the sentence.

**Q7:** *Page 3, Line 27: could you show on Fig. 3 also these relations?*

**Response:** As recommended by the reviewer, we will show the domains or relations with figures below. In the figures, the blue domain is rain domain suggested by Straka et al. (2000).

[Figure]

| (a) Z - ZDR relation | (b) Z - KDP realtion |
|---|---|

**Q8:** *Page 4, Line 8: why only positive magnitudes are considered?*

**Response:** Since YIT Radar has been installed in late 2014, it is still in the process of calibrating and optimizing the polarimetric variables for the YIT Radar measurements. The radar rainfall estimated by polarimetric variables from YIT Radar has been underestimated in the most precipitation events. We, therefore, considered only positive magnitude in the manuscript. Naturally, if the radar rainfall is overestimated, we can consider the negative magnitude. But we don't still have the cases for the overestimation of the radar rainfall in our radar.

**Q9:** *Page 4, Lines 13-15: this sentence needs some visual support (a figure), to guide the reader to understand the algorithm.*

**Response:** As recommended by the reviewer, we will explain the sentence by giving such as below figure.

[Figure]

(a)         (b)         (c)

[Figure]

|       |       |       |
|-------|-------|-------|
| (d)   | (e)   | (f)   |

Fig. Bivariate distribution of $Z$ - $Z_{DR}$ with respect to the magnitude of adjustment in Event 4: (a) no adjustment, (b) magnitude of 2, (c) magnitude of 4, (d) magnitude of 6, (e) magnitude of 8, (f) magnitude of 10

**Q10:** *Page 4, Line 18 (Table 4): add references for all the relations in the table.*

**Response:** As recommended by the reviewer, we will add below references in the Table 4.

➔ R1 : Marshall and Palmer (1948), R2 : Bringi and Chandraseker (2001), R3 : Brandes et al. (2003), R4 : WRC (2014), R5 : Cifelli et al. (2011), R6 : Ryzhkov et al. (2003)

**Q11:** *Page 5, Line 18: You should comment about those 2 behaviours (maximum around 5 dB of events 1 and 3 vs asymptotic behaviour of events 2 and 4), and here it would be helpful to understand if the type of rainfall was very different in those cases.*

**Response:** As recommended by the reviewer, we will enhance the explanation about Figure 5.

**Q12:** *Page 6, Line 25 (Fig 7): could you specify which $K_{dp}$ estimation method you employ? Sometimes the estimates of $K_{dp}$ seem poor (as in Event 1)*

**Response:** We will explain Kdp estimation method which is least square method. However Kdp in event 1 was noisy because event 1 was mostly stratiform precipitation and the rainfall intensity was not large.

**Tables and Figures:**

**Q1:** *Provide more complete information in the caption of Figures and Tables. Captions are often too short and not complete.*

**Response:** We do agree with this comment and will enhance the explanation about the caption of Figures and Tables.

**Q2:** *Table 2: it is a good starting point, but the description of the events should be more detailed and supported by actual radar images (PPI or CAPPI) for each event.*

**Response:** Please refer our response to the question for Section 2.

**Q3:** *Table 4: add a reference for the algorithms, in the same table.*

**Response:** We will add references in the Table 4.

**Q4:** *Figure 2: i like this figure, but it needs to be explained step by step with additional details in the text.*

**Response:** We have been working on enhancing interpretation of the Figure 2.

**Q5:** *Figure 4: add an indication of vertical distance between the radar measurement and the gauge.*

**Response:** This will be explained as part of the section 'DATA' in relation with the Figure 4.

---

## Author Comment (AC2) · 26 Apr 2016

**Responses to the Reviewer's Comments**

**Reviewer #2:**

We sincerely thank you for the comments that help to improve our paper. The responses to the comments are as follow.

**Comment:**

The authors suggest tweaking the measured values of Z, Zdr, and Kdp to match the average expected dependencies of Zdr and Kdp on Z or the bivariate distributions obtained from the disdrometer-based simulations. The "reference" dependencies are specified in Eqs 1 and 2. The major problem with such approach is that there are no universal reference dependencies valid for all rain types. For example, the Z –Zdr average dependency for tropical rain generated by a warm rain process is quite different from the one for continental rain which mostly originates from the ice aloft. For a given Z, Zdr in tropical rain is significantly lower than in continental rain, particularly at higher rain rates. A similar rule holds for the Z - Kdp dependency. In fact, using the suggested methodology, the authors deny the impact of the DSD variability on the performance of radar rainfall algorithms. I guess that the improvement in the QPE performance caused by the recommended adjustment is mainly due to mitigation of the measurement biases in Z and Zdr. The description of the adjustment routine in section 3.2 is very brief and insufficient for understanding or reproducing the methodology. The adjustment of Zdr or Kdp for a given Z looks straightforward but the procedure for Z adjustment is totally unclear. Obvious underestimation of rainfall, say, by using the R(Z) relation illustrated in Fig. 6a could be caused by either negative bias in the Z measurements or by the very nature of the observed rain (e.g., tropical) for which a power-law R(Z) relation with higher intercept is required. How to distinguish between these two sources of error? A range of needed adjustment (likely attributed to negative Z bias) between 3 and 10 dB shown in Table 5 is quite disturbing because it may point to a serious problem with radar calibration. The magnitude of such bias is too high for operational weather radars. Moreover, the magnitude of the Z adjustment for a single rain event can vary by as much as 3 dB depending on the algorithm choice. To me this is an indication that both Z bias and the DSD variability (which differently affects the performance of various rainfall relations) may come into play. The English usage has to be improved dramatically since even the meaning of several sentences is "lost in translation". There is inaccurate

statement regarding the methodology of Seliga and Bringi for DSD retrieval and rainfall estimation (first paragraph in Introduction). It is not a single Zdr but the combination of Z and Zdr which was proposed to address these problems.

**Response:**

Due to the introduction of dual-pol. radars, additional polarimetric variables are used for various applications in meteorological field, e.g. hydrometeor classification, detection and correction of bright band, removal of non-precipitation echo, etc. In hydrology, it is important to quantitatively estimate radar rainfall using those variables, because the radar rainfall is one of the most important input data for rainfall-runoff analysis. It is, however, more complicated to estimate rainfall of dual-pol. radar, because multivariate analysis (Z, ZDR and KDP) have to be considered. Naturally, if dual-pol radars can provide polarimetric variables without errors (bias and random error), there will be no problem at all to estimate radar rainfall. The variables, however, are affected by calibration of radar hardware, environmental interferences and rainfall events and so on. Also, errors in variables exist inevitably. For this reason, some researchers suggested methods to solve the problem such as self-consistency and ZDR calibration. Nevertheless, those methods were not able to satisfy the need of quantitatively estimating radar rainfall. We thought the empirical method in this study can contribute to solve the problem for QPE of radar.

The polarimetric variables were adjusted by using Z - ZDR relation and Z - KDP relation regardless of precipitation types. We agree with your comment that the relations will be different with various rain types. Also the reviewer commented poor calibration of the radar used in this study because of large bias in the reflectivity observed by the YIT radar. The YIT Radar was installed in August 2015 and is still in the process of calibrating and optimizing to produce more reliable data. Although the YIT radar has shown lower values for the reflectivity, the correlation coefficient compared to the gauge rainfall was greater than 0.8 even without adjustment suggested in this study. In addition atmospheric phenomena, such as bright band, are also well observed as illustrated for the Event 1 in the 'Supplementary information for the four events' later on this response. Therefore we would like to say that the YIT radar is not poorly calibrated but work in progress.

The empirical method in this study is designed not only to improve radar QPE but also to find errors in the polarimetric variables compared to raingauge measurement. The most advantage of the empirical method might be able to generalize or quantify errors including bias and random errors, which were observed or existed on radar data over long periods. It is also very important to removing bias of the polarimetric variables in order to improve the accuracy of radar QPE. However random biases or errors are not easy to remove, rather they are inevitable. Nevertheless it will be very useful to quantify the errors particularly for the better QPE, because the errors in the polarimetric variables can propagate biases in the radar rainfall estimation throughout the processes of radar QPE. Therefore the provisional aims of the empirical method are quantifying errors in radar data and improving QPE. In order to achieve these aims, we need a reference such as the relation between the polarimetric variables, Z - ZDR relation and Z - KDP relation. Of course the relations will be different depending on rain types. As a preliminary study, we apply the relations regardless of the rain types, although random errors will be inevitably occurred. It was planned to quantify random errors later when more radar data are available. At this point the section, Methodology, could was inadequate to give clear idea on the empirical method for readers; we will enhance the section in our revised manuscript.

All the authors are well aware of that our manuscript is required professional English review, and our revised manuscript will be getting a professional English correction before it will be submitted. Hope this process will improve our manuscript to reach high standard of English requirement for the journal. Also we will correct the sentence about the reference in introduction, Seliga and Bringi.

(In addition we put 'Supplementary information for the four events' to show the data used for this study, which would be helpful to relieve your concern on our radar calibration and poor quality of observation).

**Supplementary information for the four events:**

**Event 1:**

Event 1 was related to the Typhoon Chan-Hom, which was developed near the Equator, traveled through West Sea of Korea and finally hit mainland China (Fig. A). Korea had light or moderate rain over the most part of the country and rain was lasted over 24 hours since late 11th July 2015(Fig. B). Observed hourly maximum rainfall was 18mm at 201507120900KST during the event. Fig. C is shown CAPPI reflectivity image composed at 1.5 km in height using data observed by the YIT radar at 9am on 12th July 2015. Black circle is represented 100km in horizontal distance from the YIT radar and only inner circled areas are used for this study. The precipitation type was mainly stratiform rain with very clearly observed bright band as supportive evidences of Fig. D(Z), E(.  $\rho_{hv}$ ) and F(ZDR) observed by the YIT radar at 9.12°. Bright band was developed about 4.5km in height, therefore the used CAPPI image composed at 1.5km was not influenced by bright band. The YIT radar has been purposely set a beam blockage area around 0~10degree to prevent intervenes by neighboring telecommunication radar.